# Genome-Wide Identification and Analysis of the Growth-Regulating Factor Family in *Zanthoxylum armatum* DC and Functional Analysis of *ZaGRF6* in Leaf Size and Longevity Regulation

**DOI:** 10.3390/ijms23169043

**Published:** 2022-08-12

**Authors:** Yanhui Huang, Jiajia Chen, Jianrong Li, Yan Li, Xiaofang Zeng

**Affiliations:** Key Laboratory of Plant Resource Conservation and Germplasm Innovation in Mountainous Region (Ministry of Education), College of Life Sciences/Institute of Agro-Bioengineering, Guizhou University, Guiyang 550025, China

**Keywords:** growth-regulating factor, *Zanthoxylum armatum*, *ZaGRF6*, growth, leaf senescence

## Abstract

Growth-regulating factors (GRFs) are plant-specific transcription factors that play an important role in plant growth and development. In this study, fifteen *GRF* gene members containing QLQ and WRC domains were identified in *Zanthoxylum armatum*. Phylogenetic and collinearity analysis showed that ZaGRFs were closely related to CsGRFs and AtGRFs, and distantly related to OsGRFs. There are a large number of cis-acting elements related to hormone response and stress induction in the *GRF* gene promoter region of *Z. armatum*. Tissue-specific expression analysis showed that except for *ZaGRF7*, all the *ZaGRFs* were highly expressed in young parts with active growth and development, including terminal buds, seeds, and young flowers, suggesting their key roles in *Z. armatum* growth and development. Eight *ZaGRFs* were selected to investigate the transcriptional response to auxin, gibberellin and drought treatments. A total of six *ZaGRFs* in the NAA treatment, four *ZaGRFs* in the GA_3_ treatment, and six *ZaGRFs* in the PEG treatment were induced and significantly up-regulated. Overexpression of *ZaGRF6* increased branching and chlorophyll content and delayed senescence of transgenic *Nicotiana benthamiana*. *ZaGRF6* increased the expression of *CRF2* and suppressed the expression of *ARR4* and *CKX1*, indicating that *ZaGRF6* is involved in cytokinin metabolism and signal transduction. These research results lay a foundation for further analysis of the *GRF* gene function of *Z. armatum* and provide candidate genes for growth, development, and stress resistance breeding of *Z. armatum*.

## 1. Introduction

Growth-regulating factors (GRFs) are plant-specific transcription factors that play an important role in growth and response to abiotic stress [1,2,3]. The GRF protein contains two conserved domains, QLQ (Gln, Leu, Gln) and WRC (Trp, Arg, Cys) [1]. The QLQ domain interacts with GRF interaction factor (GIF) to form a transcription co-activator complex [4]. The WRC domain is the DNA-binding region of the GRF protein, which consists of a nuclear localization signal motif and a DNA-binding zinc finger structure [5]. The first member of the identified *GRF* gene is *OsGRF1*, which plays a regulatory role in gibberellin (GA_3_)-induced stem elongation [1]. Studies have shown that *OsGRF1* has multiple functions. The *OsGRF1* gene is involved in regulating growth at the juvenile stage, and may be involved in the regulation of heading in rice as well [1,6]. There is a close relationship between plant hormones and GRFs in plant growth and development. In tobacco, twelve *NtGRFs* were enhanced by GA_3_ treatment [7]. Exogenous GA_3_ application activated *AhGRF5* and showed a more positive response [8]. In apple, exogenous 1-naphthylacetic acid (NAA) and GA_3_ can elevate the expression level of five and seven *MdGRF* genes, respectively [9]. In *Arabidopsis*, *AtGRF5* stimulates chloroplast division, leaf expansion and longevity [10]. GRFs have been reported to coordinate plant growth with cytokinin to affect plant senescence. The suppressive role of GRFs in leaf senescence may be explained by the GRF-cytokinin interaction, as 35S:AtGRF5 increases the sensitivity of leaves to cytokinins. Studies have shown that GRFs can respond positively to drought stress, suggesting that GRFs play an important role in plant response to abiotic stress. The overexpression of *AtGRF7* results in upregulation of *DREB2A*, which confers increased tolerance to salt and drought stress [11]. *MsGRF2* and *MsGRF6* were significantly up-regulated under mannitol treatment in *Medicago*, indicating that they are most likely involved in drought stress tolerance [12]. To date, GRFs have been identified in multiple species. Among them are nine GRFs in *Arabidopsis* [5], thirteen in tobacco [7], twelve in rice [6], sixteen in apple [9], and thirty in wheat [13].

*Zanthoxylu armatum* DC, an important aromatic woody shrub in Rutaceae, has a long history of usage and cultivation as an important spice and medicinal plant in Asia [14,15]. The fruit of *Zanthoxylum* is important as a seasoning and for sesame oil, with good edible value [16]. Due to its high tolerance to drought and calcareous soils, *Zanthoxylum* is an economically valuable plant, grows well under moderately arid to semi-arid conditions, and has the potential to restore degraded land [17]. At present, most of the research on *Zanthoxylum* has focused on its pharmacological and antioxidant activity, while the regulatory mechanism of its growth and development and related gene functions are remain less studied. Although *GRF* family members of many plants have been identified and studied, studies on *GRF* genes in *Z. armatum* are not yet available. The release of the whole genome data of *Z. armatum* is of great significance for understanding its biological characteristics [14]. Therefore, identifying and characterizing *GRF* genes in *Z. armatum* is of great interest.

In this study, fifteen *ZaGRF* gene family members were identified from *Z. armatum*. The structural characteristics, phylogenetic relationships, chromosome localizations, synteny analysis, and expression patterns of the *ZaGRF* gene family were analyzed. In addition, the expression profiles of *ZaGRF* genes in response to hormone and drought treatment were ascertained. Furthermore, we identified a possible functional gene, *ZaGRF6,* involved in cytokinin metabolism and signal transduction. Genome-wide identification and expression analysis of the *ZaGRF* gene family provides a basis for further clarifying the important functions of *GRF* genes in the growth and development of *Z. armatum*.

## 2. Results

### 2.1. Identification of the GRF Gene Family in Z. armatum

Via Hidden Markov Model searches, twenty *GRF* genes were identified from the *Z. armatum* genome. Finally, fifteen *GRF* genes were obtained by NCBI-CDD and SMART and the *ZaGRFs* were designated as *ZaGRF1* to *ZaGRF15* based on their positions on their distribution in the *Z. armatum* genome. The deduced polypeptides ranged in length from 272 (ZaGRF13) to 652 (ZaGRF8) amino acids, with molecular weights between 31.14 kD (ZaGRF5) and 70.82 kD (ZaGRF8). The isoelectric points ranged from 6.24 (ZaGRF8) to 9.90 (ZaGRF7). The instability coefficient results showed that all ZaGRFs were unstable proteins (instability coefficient > 40) (Table 1).

The multiple sequence alignment indicated that all of the putative ZaGRF proteins have both QLQ and WRC domains in the N-terminal region (Figure 1). The N-terminal QLQ domain was conserved with one Leu and two Gln residues in all the ZaGRF proteins except for ZaGRF2. The WRC domain was highly conserved, with one Trp, Arg, and Cys in each of the ZaGRF proteins. The zinc finger structure (CCCH) was found within the WRC domain in all ZaGRF proteins except for ZaGRF12, ZaGRF14, and ZaGRF15 (Figure 1).

MiRNAs typically recognize their target mRNAs through partial complementarity [18]. Previous studies have found that GRF genes are the targets of miR396. MiR396 directly targets *GRF* transcripts, thereby negatively regulating gene expression levels [19]. Here, sRNATarget targeting software was used to predict the target sites of miR396 in *ZaGRFs*. The results showed that a total of thirteen *ZaGRFs*, excepting *ZaGRF2* and *ZaGRF5*, contained target sites for miR396. Therefore, we speculated that these thirteen genes might be the target genes of miR396 (Figure 2).

### 2.2. The Protein Structure, Subcellular Localization Prediction, and Post-Translational Modification of ZaGRFs

The protein subcellular localization of ZaGRF was analyzed by Protcomp. The results showed that eleven members of ZaGRF were presumably localized in the nucleus, while four members (ZaGRF2, ZaGRF12, ZaGRF14, and ZaGRF15) were presumably localized in the extracellular matrix. The secondary structure prediction results showed that the ZaGRF proteins contained three types of secondary structures, namely, α-helix, extended chain, and random coil, of which random coil accounted for the largest proportion at 66.58% on average (Appendix A). The results of the 3D structure showed that ZaGRFs had great similarities in the spatial conformation of protein three-dimensional structures, all containing a large number of irregular curls, which is essentially the same as the secondary structure prediction structure (Figure 3a).

The post-translational modifications of ZaGRFs were predicted in terms of phosphorylation. A total of 924 potential phosphorylation events on the serine, threonine, and tyrosine were identified within ZaGRFs (Figure 2b). The phosphorylation events were predicted to be related to serine (666), followed by threonine (191), and then by tyrosine (67). Among the ZaGRFs, most of the phosphorylation sites (99 sites) were predicted in ZaGRF8, whereas the phosphorylation events ranged from 44 to 81 sites in other ZaGRFs.

### 2.3. Phylogenetic Analysis

The phylogenetic analysis of 43 GRF-proteins from *Z. armatum* (15), *O. sativa* (12), *C. sinensis* (7) and *A. thaliana* (9) were divided into seven groups (Figure 4 and Appendix A). The ZaGRFs were distributed into four groups (Group A to D). Group A included only one protein, ZaGRF2. Group C was confined to two sequences, ZaGRF3 and ZaGRF8. Group D consisted of five ZaGRFs, ZaGRF1, ZaGRF5, ZaGRF6, ZaGRF7, and ZaGRF13. Seven ZaGRFs (ZaGRF4, ZaGRF9, ZaGRF10, ZaGRF11, ZaGRF12, ZaGRF14, and ZaGRF15) were clustered in Group B.

### 2.4. Gene Structure, Motif Distribution, and Conserved Domain Alignment

To understand the gene structure diversity and conserved motifs of the *Z. armatum* GRF protein, the intron–exon regions in genes and motifs in proteins were systematically analyzed. Gene structure diagram analysis showed that fifteen *ZaGRFs* had various numbers of exons, ranging from three to six (Figure 5a). Most *ZaGRF* genes contained four exons. Among them, the number of exons of *ZaGRF3* and *ZaGRF8* were at most six, while for *ZaGRF7* and *ZaGRF11* there were at least two. Ten motifs were revealed in *ZaGRFs* (Figure 5b,c). Conserved motif analysis showed that all fifteen members contained Motify1 (WRC) and Motify2 (QLQ) (Figure 5b,c). The motifs of proteins that clustered together within the phylogenetic tree presented similarities in the distribution of motifs to an extent (Figure 4b).

### 2.5. Chromosome Localization and Synteny Analysis

Eight genes (*ZaGRF1-ZaGRF8*) were anchored on eight chromosomes, while *ZaGRF9-ZaGRF15* were not anchored on chromosome. The chromosomal location map of *ZaGRF* genes showed that *ZaGRF1-ZaGRF8*, were anchored on chromosomes 11, 15, 18, 20, 21, 26, 27, and 31, respectively. Chromosome 27 contained two *ZaGRF* genes (*ZaGRF7* and *Zardc38469.t1*), and *ZaGRF3* and *ZaGRF8* were segmental duplication genes (Figure 6a). The collinear relationship between *Z. armatum* and *C. sinensis* and *A. thaliana* was significantly greater than that of *O. sativa*. Five *ZaGRF genes* shared homology with *C. sinensis*, seven *ZaGRF* genes shared homology with *A. thaliana*, and only one *ZaGRF* gene shared homology with *O. sativa* (Figure 6b). This may be related to the fact that *Z. armatum*, *C. sinensis*, and *A. thaliana* are all dicotyledonous plants, and *C. sinensis* belongs to *Rutaceae*, which is closely related.

### 2.6. Promoter Analysis

Surveying *cis*-elements in promoter regions can help in better understanding the potential function and regulatory mechanism of the *GRF* gene in *Z. armatum*. The analyses of cis-regulatory elements in promoter regions revealed the presence of binding sites for key transcription factors related to CAAT-box and TATA-box elements, light-responsive elements, hormone-responsive elements, stress-related elements, and growth-response elements (Appendix A). Among them, cis-elements related to the hormone response include ARE (auxin), ABRE (abscisic acid), CGTCA-motif (methyl jasmonate), and P-box (gibberellin). Elements related to the stress response include STRE (stress response), LTR (low temperature), ACE (anaerobic induction), TC-rich repeats (defense and stress), MBS (MYB binding site involved in drought), and WUN-motif (mechanical injury). The O2-site (zein metabolic regulatory element) is related to growth and development (Appendix A).

### 2.7. In Silico Tissue-Specific Expression of ZaGRF Genes

To further explore the function of *ZaGRFs*, we analyzed the expression levels of *ZaGRFs* in nine tissues: young leaf, mature leaf, petiole, terminal bud, stem, young flower, prick, seed, and husk. RNA sequencing (RNA-Seq) data were downloaded from the SRA database published by Wang et al. [14]. Differential expression was noted for *ZaGRF* genes in various tissues of *Z. armatum*. In general, the *ZaGRF* genes were constitutively expressed in all tested tissues except for *ZaGRF5* and *ZaGRF11*. *ZaGRF5* was only expressed in seeds, young flowers, and terminal buds, while *ZaGRF11* was not expressed in leaves. Except for *ZaGRF7*, all *ZaGRF* genes were expressed at higher levels in terminal buds, seeds, and young flowers than in other tissues, suggesting that these genes play an important role in *Z. armatum* growth and development. Among them, the expression levels of *ZaGRF3* and *ZaGRF8* in these three tissues were significantly higher than those of other genes, and *ZaGRF1* had the highest expression in leaf and husk [5]. In addition, *ZaGRF13* was highly expressed in young flowers, and *ZaGRF9* was highly expressed in seeds (Figure 7). These expression results showed that *ZaGRF* family members may play important roles in the development of *Z. armatum* shoot tip meristems, seeds, and flowers.

### 2.8. ZaGRF Gene Expression Profiles in Response to Hormone and Drought Treatment

Previous studies have demonstrated that GRFs play important roles in plant responses to hormone and abiotic stresses. Our cis-element analysis showed that there are a large number of hormone-responsive and drought- responsive elements in the *ZaGRF* gene promoter region. We selected eight *ZaGRFs* for qRT-PCR to investigate the transcriptional response to auxin, gibberellin and drought treatments. Comprehensive expression profiles of *ZaGRF* genes under hormone and drought treatments are shown in Figure 8. The results show that most of the *ZaGRFs* exhibited significantly altered transcript levels after treatment. A total of six *ZaGRFs* in the NAA treatment, four *ZaGRFs* in the GA_3_ treatment, and six *ZaGRFs* in the PEG treatment were up-regulated in transcription by two-fold to 64.9-fold. The highest-fold inductions in the transcriptional responses to hormones and drought were exhibited by *ZaGRF3* (64.9-fold to NAA), *ZaGRF6* (13.3-fold to GA_3_) and *ZaGRF8* (9.1-fold to PEG). Notably, *ZaGRF1* and *ZaGRF3* accumulated higher transcription levels in response to all the treatments. *ZaGRF6* was strongly induced by GA_3_ and responded only slightly to PEG. The expression of *ZaGRF4* and *ZaGRF5* was suppressed by GA_3_ and PEG, and *ZaGRF4* was not sensitive to NAA. *ZaGRF7* was inhibited by GA_3_ and up-regulated by NAA treatment. These results indicate that *ZaGRF* genes likely function in a manner that is responsive to hormone signal transduction and drought stress response.

### 2.9. Overexpression of ZaGRF6 Alters Leaf Size and Longevity in Transgenic Nicotiana Benthamiana

To further investigate the function of *ZaGRF6*, the overexpression construct 35S:ZaGRF6-GFP was transformed into *Nicotiana benthamiana*. Thirty-two independent transgenic lines were obtained. The expression levels of *ZaGRF6* in transgenic lines ZG1 and ZG2 were detected, then the two lines were used for further analysis. The overexpression transgenic lines showed various physiological and morphological abnormalities, including shorter petioles, thicker and smaller leaves, darker green leaf color, and increased branch number compared with the wild type (Figure 9a,b). A previous study showed that *GRF5* could enhance chlorophyll retention after dark-induced senescence [10], indicating that leaf senescence is postponed by cytokinins, which could serve as an additional commonality between *GRF5* and cytokinin functions. Leaves from the transgenic lines and WT were kept in darkness in vitro to observe their senescence responses. Compared with the WT, the transgenic lines showed delayed leaf senescence (Figure 9c). The chlorophyll (Chl) content was significantly higher in transgenic leaves than in wild-type leaves under normal greenhouse conditions (Figure 9d). The leaves of WT plants started to turn yellow after 5 d in the dark, and completely turned yellow and withered after 12 d. After 5 d in the dark, the leaves retained 75% of the total Chl that was present before incubation, and the leaves retained only 45% after 9 d. The leaves of ZG1 began to turn yellow after 9 d in the dark, and the Chl retention was 70%. However, the leaves of ZG2 showed no significant change in leaf color after 12 d, and the leaf Chl was retained at 80% (Figure 9e). Furthermore, we detected the expression levels of *ZaGRF6*, *PORB*, *GLK1*, *ARR4*, *CRF2*, and *CKX1* in transgenic plants. The results showed the expression level of *ZaGRF6* in ZG2 was 32-fold higher than that in ZG1, and the expression levels of *GLK1*, *ARR4*, and *CKX1* in transgenic plants were significantly decreased compared with those in wild type plants (Figure 10). The expression level of *CRF2* was up-regulated 1.5- and 2.2-fold in ZG1 and ZG2, respectively. There was no obvious difference in the expression of *PORB* (Figure 10). The leaves of *ZaGRF6*-overexpressing plants showed more intense greening and Chl retention, suggesting an increase in Chl levels and senescence delay, which could be caused by altered cytokinin signaling.

## 3. Discussion

GRFs are a class of transcription factors that are widely involved in leaf and seed development, pistil development, flower regulation, abiotic stress responses, and exogenous hormone responses [4,10,11,20,21,22]. In the present study, we identified fifteen *GRF* genes from *Zanthoxylum armatum* genome. The structural features of ZaGRF proteins contained QLQ and WRC domains, which are similar to those of the GRF proteins of Arabidopsis and rice [5,6]. Although the Gln-Leu-Gln residues of the QLQ domain are absolutely conserved in ZaGRFs, this feature is absent in ZaGRF2 due to the substitution of Leu by Phe. It is well known that transcription factors act in the nucleus to regulate the expression of target genes. The results of subcellular localization prediction showed that eleven ZaGRFs were located in the nucleus, suggesting that most ZaGRFs are subject to normal transcription. Protein phosphorylation is one of the most common and important post-translational modifications, and occurs as a mechanism to regulate the biological activity of a protein [23]. We predicted phosphorylation sites in all ZaGRFs, ranging in number from 44 to 99. The phylogenetic analysis showed that ZaGRFs clustered preferentially with CsGRFs and AtGRFs, indicating that *Z. armatum* is closely related to *C. sinensis* and *Arabidopsis* and distantly related to the monocotyledon rice. The collinearity analysis showed that there was a strong linear homologous relationship between *Z. armatum* and *C. sinensis*, followed by *Arabidopsis* and rice. Collinear analysis showed that only two pairs of segmental duplication genes (*ZaGRF3*:*ZaGRF8*, *ZaGRF7*:Zardc38469.t1) were found in *Z. armatum*. We found that Zardc38469.t1 only contains the WRC domain, not the QLQ domain. This suggests that fewer gene duplication events have occurred in the *ZaGRF* family during evolution. Synteny analysis showed that seven pairs of duplication genes were found in *Z. armatum* and *Arabidopsis* and five pairs of duplication genes in *Z. armatum* and *C. sinensis* (Appendix A). However, only one pair of duplication genes was found between *Z. armatum* and rice (*ZaGRF5*: *OsGRF1*) (Appendix A). Consistent with the phylogenetic tree, the collinearity analysis revealed that *ZaGRF3*, *ZaGRF8*, *AtGRF1*, *AtGRF2*, and *CsGRF2* probably share the same ancestor. These results demonstrate that *ZaGRF5* completed duplication before the separation and evolution of monocotyledonous and dicotyledonous plants. Therefore, *ZaGRF5*, *AtGRF5*, *AtGRF6*, and *OsGRF1* might originate from the same ancient ancestor.

Previous studies have indicated that the *GRF* gene family is involved in hormone signal transduction and abiotic stress responses. There are several cis-acting elements related to hormone response and stress induction in the promoter region of the *GRF* genes of *Z. armatum*. Our study found that most of the *ZaGRFs* were induced by hormone and drought stress treatments. Under NAA treatment, the expression levels of most *ZaGRFs* were significantly upregulated. Among them, the expression of *ZaGRF3* increased 65-fold compared to that before treatment, whereas *ZaGRF4* and *ZaGRF5* were downregulated. These results suggest that different genes may perform different functions in the same gene family [24]. In rice, GA_3_ can suppress the expression of miR396, thereby upregulating the transcript level of *OsGRF6* [25]. In our study, under GA_3_ treatment, the transcript levels of five predicted miR396 target genes (*ZaGRF1*, *ZaGRF3*, *ZaGRF6*, *ZaGRF8* and *ZaGRF15*) were significantly upregulated. It has been reported that GRF transcription factors play key roles in plant growth by coordinating stress responses and defense signals [11,26,27]. In several studies, drought treatment has been found to increase the expression level of GRFs [13,28]. In the present study, we found that six *ZaGRF* genes were upregulated under PEG treatment, especially *ZaGRF1* and *ZaGRF8*. This shows that *ZaGRF1* and *ZaGRF8* may play important roles in the responses to abiotic stress in *Z. armatum*. The expression levels of *ZaGRFs* peaked 24 h to 48 h after PEG6000 treatment. We speculate that the slow response of *ZaGRF* genes to drought stress suggests that these genes are not upstream of the osmotic stress signaling pathway [13].

GRFs generally play positive roles in plant growth and development. Studies have shown that overexpression of GRFs leads to increase cell proliferation and leaf expansion in rice, *Arabidopsis*, poplar, and lettuce [10,21,29,30,31]. However, *GRF* genes have been reported to play negative roles in plant growth as well. Overexpression of *ZmGRF10* resulted in a reduction in leaf size and plant height [20]. *AtGRF9* negatively regulates Arabidopsis leaf growth by restricting cell proliferation in leaf primordia. The *grf9* mutants had larger rosette leaves and petals than WT, while plants overexpressing *AtGRF9* produced smaller leaves and petals [24]. In rice, overexpression of *OsGRF7* causes a semi-dwarf and compact plant architecture with increased culm wall thickness and narrowed leaf angles [32]. In this study, plants overexpressing *ZaGRF6* showed smaller leaves and shorter petioles, indicating that *ZaGRF6* plays a negative role in regulating *N. benthamiana* leaf development. However, in addition to the smaller leaves, transgenic plants showed cytokinin-overproducing phenotypes, such as increased chlorophyll content and branching and delayed leaf senescence [33,34]. It is suggested that the regulatory mechanisms of *ZaGRF6* and these GRFs that have opposite effects on growth and development may be different.

Cytokinins generally function in promoting mitotic cell division [35], regulating shoot apical meristem (SAM) activity and organizing shoot architecture and leaf senescence [36]. Preventing decline in cytokinin levels by expressing the *ipt* gene has been shown to delay the senescence of tobacco and *Zanthoxylum* [33,34]. In our study, the transgenic lines ZG1 and ZG2 showed darker green leaves with higher chlorophyll contents than the wild type. Delayed leaf senescence was found in the transgenic plants overexpressing *ZaGRF6*. The higher *ZaGRF6* expression level caused a more severe phenotype in the transgenic line. These results suggest that there may be a relationship between *ZaGRF6* and cytokinin. Currently, there is evidence that GRFs directly regulate the cytokinin degradation gene *CKX1* and are involved in cytokine signal transduction. In rice, *OsGRF4* regulates two cytokinin dehydrogenase precursor genes (*OsCKX5* and *OsCKX1*), resulting in increased cytokinin levels [21]. *PpnGRF5-1* binds to the *PpnCKX1* promoter directly and represses its expression. Overexpression of *PpnGRF5-1* repressed cytokinin degradation and significantly increased zeatin and isopentenyladenine in poplar apical buds and third leaves [30]. In Arabidopsis, *AtGRFs* interact with AtGIF1/AN3 to directly activate *AtCRF2* and repress *AtARR4* [29]. High *AtGRF3* and *AtGRF5* activity delayed leaf senescence. Evidence has demonstrated that *GRF5* and cytokinin functions are interconnected during senescence. The *AtGRF5* gene stimulates chloroplast proliferation, resulting in a higher chloroplast number per cell with a concomitant increase in chlorophyll levels in *35S:GRF5* leaves. 35S:GRF5 leaves showed enhanced sensitivity to cytokinin-driven stimulation of Chl retention after dark-induced senescence [10]. The cytokine response factor *AtCRF2* was upregulated in 35S:GRF plants. The A-type ARRs *ARR4*, *ARR5*, *ARR6*, and *ARR9* were significantly repressed in 35S:GRF5 plants [10]. In our study, the expression levels of the *CKX1* and *CRF2* genes in ZG1 and ZG were significantly increased, while the *ARR4* gene was suppressed. Evolutionary tree analysis results show that ZaGRF6 has high homology with OsGRF4, indicating that *ZaGRF6* and *OsGRF4* may have similar functions. We hypothesized that *ZaGRF6* may regulate the growth and development of *N. benthamiana* plants by participating in cytokinin metabolism and signal transduction. Further research is required to explore the relationship between the *ZaGRF6* gene and cytokinin in transgenic plants.

## 4. Materials and Methods

### 4.1. Plant Materials and Treatments

Three-year-old plants of Dingtan pepper (*Zanthoxylum armatum* var. *dintanensis*) were selected for expression pattern experiments. In this experiment, the pepper plants were irrigated with 20% PEG for drought treatment, and the young leaves were sprayed with 300 µg/L NAA and 400 µg/L GA_3_ for hormone treatments. Leaves (0.2 g) were collected at 0, 1, 2, 4, 6, 12, 24, and 48 h post-treatment. All the leaves were immediately snap-frozen in liquid nitrogen and then stored in a −80 °C ultralow temperature freezer. *Nicotiana benthamiana* was used as the wild-type (WT) control, and all transgenic lines were generated in the background of *Nicotiana benthamiana* in this study. All transgenic and WT plants were grown in a glasshouse (16 h light/8 h dark cycles; 24 °C; relative humidity 70%).

### 4.2. Protein Identification, Physicochemical Properties, and Mir396 Target Site Analysis

The protein sequence of *Z. armatum* was downloaded from the whole genome database of *Z. armatum* published by Wang, Tong, Ma, Xi and Liu [14]. The conserved GRF protein domains QLQ (PF08880) and WRC (PF08879) were used to establish an HMM model to identify homologous sequences using HMMER 3.0_Windows software (E-value 1 × 10^−5^) [37,38]. Then, the GRF family members were identified by CDD (https://www.ncbi.nlm.nih.gov/cdd, accessed on 16 September 2021) and SMART software (http://smart.embl-heidelberg.de/, accessed on 16 September 2021). The physicochemical properties of GRF family members were analyzed by the ProtParam tool of EXPASY (http://web.expasy.org/protparam, accessed on 28 September 2021). Additionally, multiple alignment of ZaGRF protein sequences was performed by DNAMAN6.0 software. The target sites of Mir396 were predicted using psRNAtarget software (http://plantgrn.noble.org/psRNATarget/, accessed on 14 January 2022).

### 4.3. Prediction of Protein Structure, Subcellular Localization, and Post-Translational Modifications

The protein secondary and three-dimensional structures were predicted by NPSA (https://npsa-prabi.ibcp.fr/, accessed on 30 September 2021) and SWISS-MODEL (https://www.swissmodel.expasy.org/interactive, accessed on 9 October 2021), respectively. The online tool Protcomp (http://www.softberry.com/berry.phtml/, accessed on 11 October 2021) was used to predict the protein subcellular localization. The phosphorylation sites of the ZaGRF proteins were predicted by the NetPhos 3.1 server with a potential value >0.5.

### 4.4. Phylogeny, Conserved Motifs, and Gene Structure Analysis

The GRF protein phylogeny among *Zanthoxylum armatum*, *Citrus sinensis*, *Arabidopsis thaliana*, and *Oryza sativa* was constructed by the Neighbor-Joining (NJ) adjacency method in MEGA11, and the bootstrap was set to 1000. The protein sequences of *A. thaliana* and *O. sativa* were downloaded from plantTFDB (http://planttfdb.gao-lab.org/index.php, accessed on 26 November 2021), and the sequences of *C. sinensis* were downloaded from Citrus sinensis v3.0 (NCBI) (http://planttfdb.gao-lab.org/index.php, accessed on 26 November 2021). The conserved motifs of the ZaGRFs were identified by the MEME program (https://meme-suite.org/meme/tools/meme, accessed on 13 December 2021). The gene structures were mapped using Gene Structure Display Server 2.0 (http://gsds.gao-lab.org/, accessed on 7 January 2022).

### 4.5. Chromosome Location and Collinearity Analysis

The chromosomal locations of the *ZaGRF* genes were determined according to the GFF3 file and mapped with TBtools software [39]. MCScanX v1.1 with default parameters in TBtools was used to analyze the collinearity among the *ZaGRFs* [40]. Then, MCScanX v1.1 and Circos 0.69.8 in TBtools were used to retrieve and map collinearity among *Z. armatum*, *C*. *sinensis*, *A. thaliana*, and *O. sativa*.

### 4.6. Cis-Acting Element and Gene Expression Pattern Analysis

The promoter sequences, 2000 bp regions upstream of the ZaGRF translational start sites were extracted from the genome sequence by SequenceToolkit in TBtools. The cis-regulatory elements of promoter sequences were analyzed by PlantCARE (http://bioinformatics.psb.ugent.be/webtools/plantcare/html/, accessed on 16 September 2021) and visualized by TBtools. Transcriptome data for nine different tissues (young leaf, mature leaf, petiole, terminal bud, stem, young flower, prick, seed, and husk) were downloaded from the NCBI database (PRJNA721257) by using the SRA Toolkit software [14]. The SRA files were converted to fastq files by FASTQ software [41]. Then, FastQC software was used for data quality control [42]. Trimmomatic was used for data filtering and removing adapters [43]. Kallisto was used for quantitative data processing [44]. The TBtool software was used for heatmap visualization.

### 4.7. Plasmid Construction and Plant Transformation

The full length CDS of *ZaGRF6* was cloned from Dingtan pepper and constructed into the overexpression vector pCambia1300-GFP by the 35S promoter. The 35S:ZaGRF6-GFP vector was transformed into *Nicotiana benthamiana* by *Agrobacterium*-mediated transformation following the leaf disc method. The roots of transgenic plants were used to investigate the subcellular localization of ZaGRF6. The green fluorescent protein (GFP) fluorescence signal was observed with a laser confocal microscope (Leica SP8 STED) under excitation at 488 nm.

### 4.8. Chl Measurements after Dark-Induced Senescence

To test the tolerance of plant leaves under dark-induced senescence conditions, leaves from WT plants and transgenic lines were detected on moist filter paper in 10-cm Petri dishes and then placed in the dark at 28 °C. The leaf chlorophyll concentration was measured by a SPAD-502PLUS meter (KONICA MINOLTA, Tokyo, Japan).

### 4.9. RNA Isolation and qRT–PCR Analysis

Total RNA was extracted by using Plant RNA Kit (Omega Bio-Tek, Doraville, GA, USA) according to the manufacturer’s instructions. The cDNAs were generated by RT-PCR using the StarScript III RT Mix Kit (GenStar, Beijing, China). The qRT-PCR analyses were performed using a qTower3G Real-time PCR System (Analytik Jena AG, Jena city, Germany) and SYBR^®^ Green Fast Mixture (GenStar, Beijing, China). The *ZaActin* and *NbActin* genes were used as internal references to normalize the gene expression levels. The relative gene expression levels were calculated using the 2^−ΔΔ*C*t^ method [45]. The primers used for real-time PCR are listed in Appendix A.

## Figures and Tables

**Figure 1 ijms-23-09043-f001:**
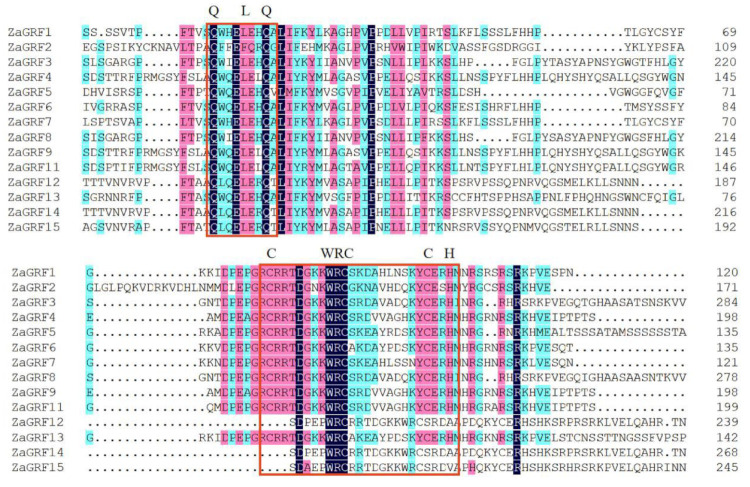
Comparison of multiple amino acid sequences of GRF members of *Z. armatum.* Different colors indicate different degrees of similarity (black: 100%; pink: > 75%; blue: > 50%).

**Figure 2 ijms-23-09043-f002:**
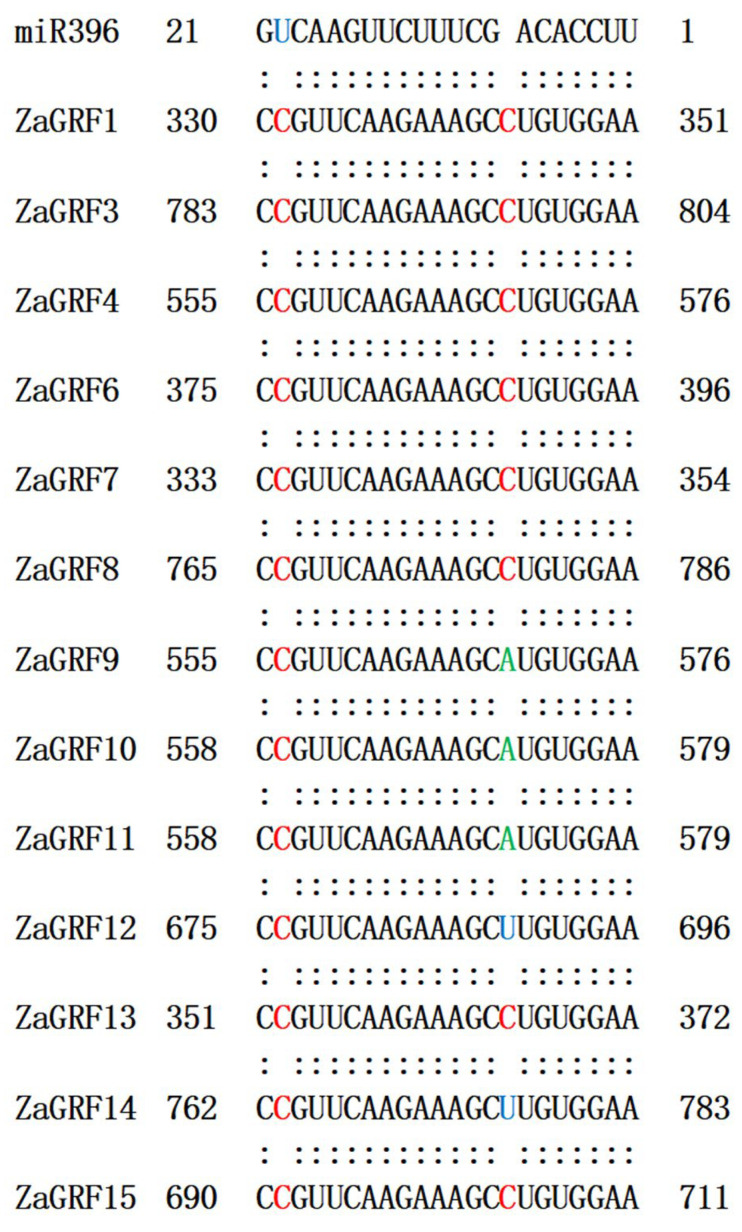
Prediction of miR396 targets of *ZaGRF* genes.

**Figure 3 ijms-23-09043-f003:**
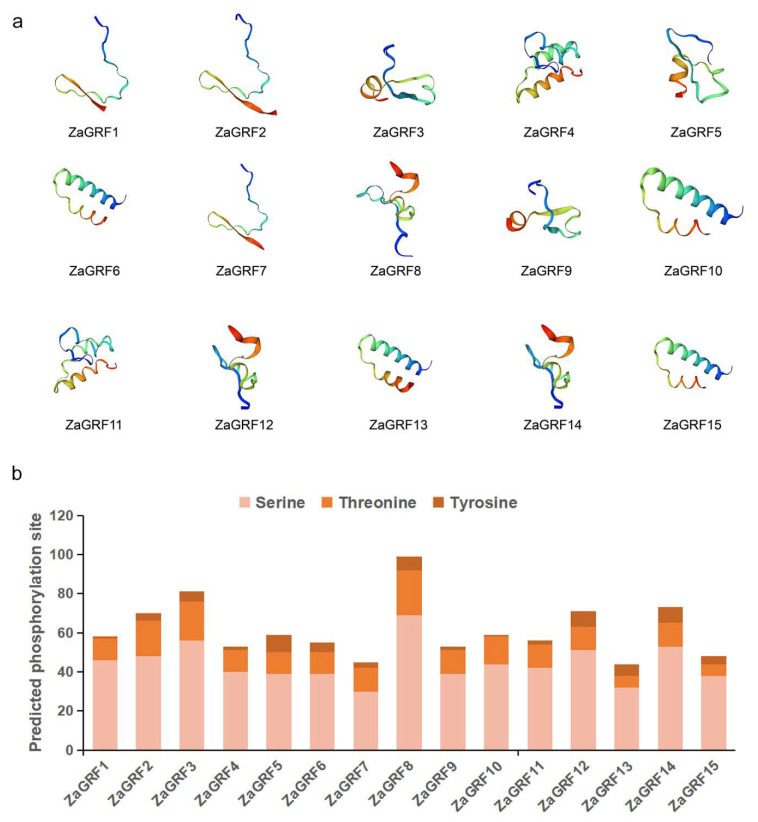
The three-dimensional structures (**a**) and phosphorylation site (**b**) of ZaGRFs.

**Figure 4 ijms-23-09043-f004:**
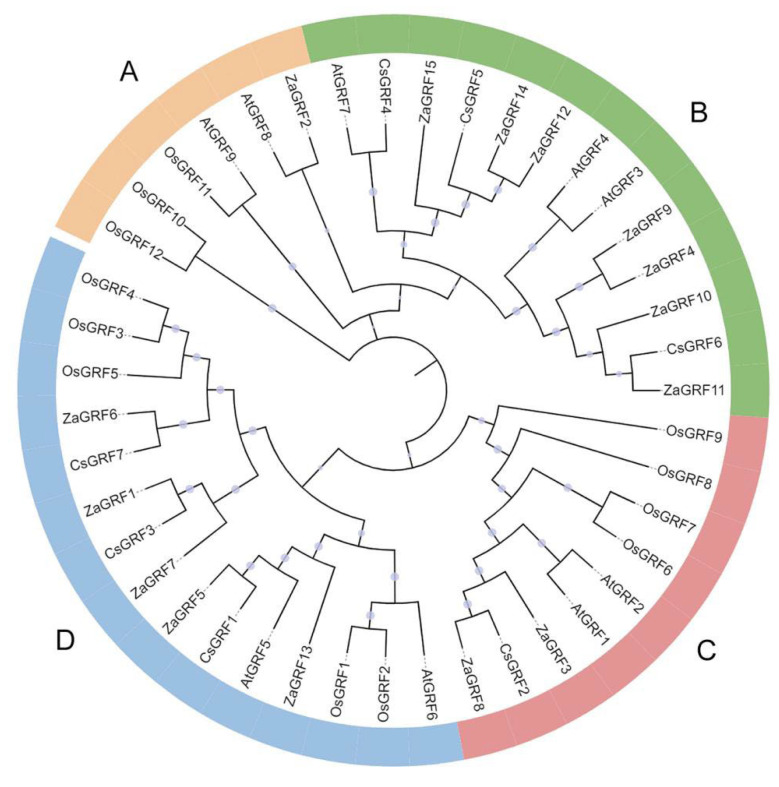
Phylogenetic relationship of ZaGRF proteins along with proteins of three other species. The start of each sequence contained codes for the following species: Za: *Zanthoxylum armatum*, Cs: *Citrus sinensis*, At: *Arabidopsis thaliana*, Os: *Oryza sativa*.

**Figure 5 ijms-23-09043-f005:**
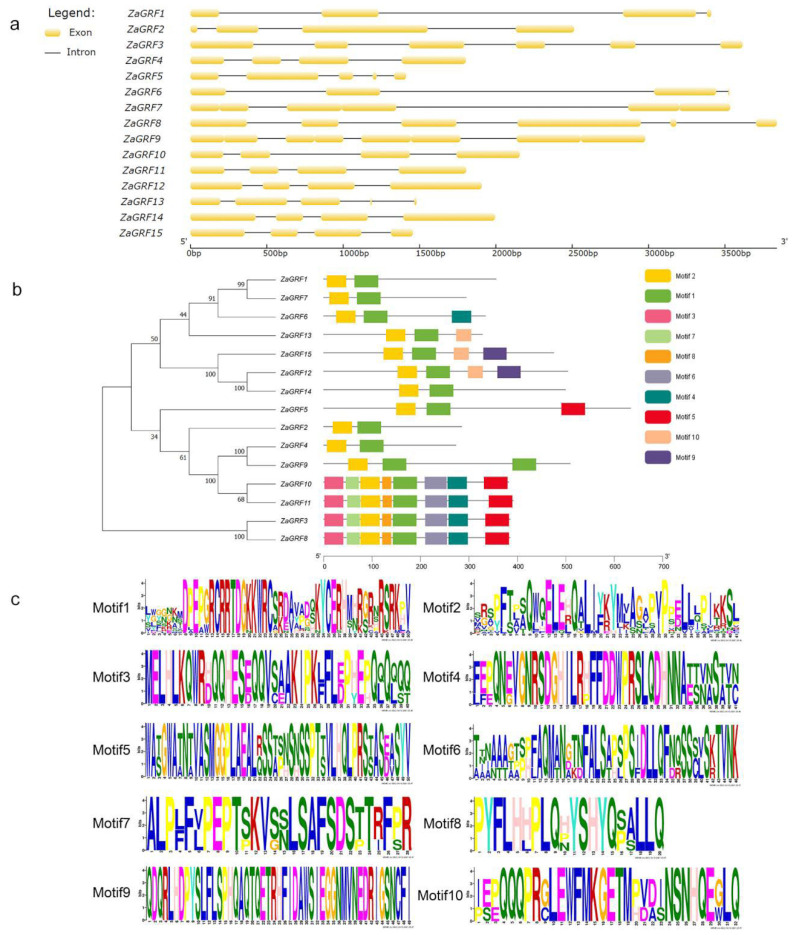
Gene structures (**a**), phylogenetic tree and motif distribution (**b**) and logo map of motifs (**c**) of *ZaGRFs*.

**Figure 6 ijms-23-09043-f006:**
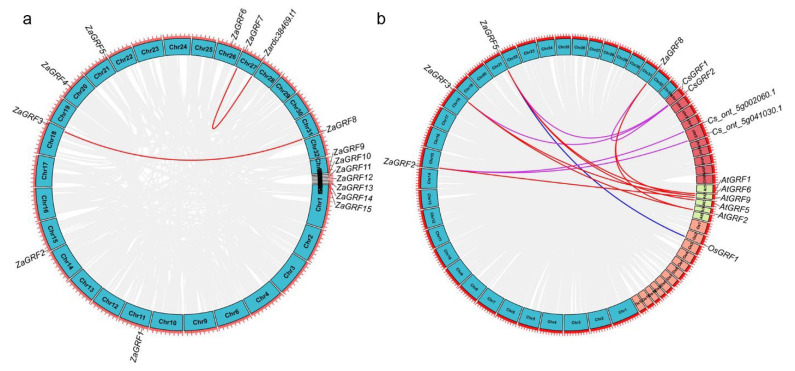
Colinear analysis. (**a**) Chromosome distribution and collinearity analysis of *ZaGRF* genes and (**b**) Colinear analysis of *GRF* genes among *Zanthoxylum armatum*, *Citrus sinensis*, *Arabidopsis thaliana* and *Oryza sativa*.

**Figure 7 ijms-23-09043-f007:**
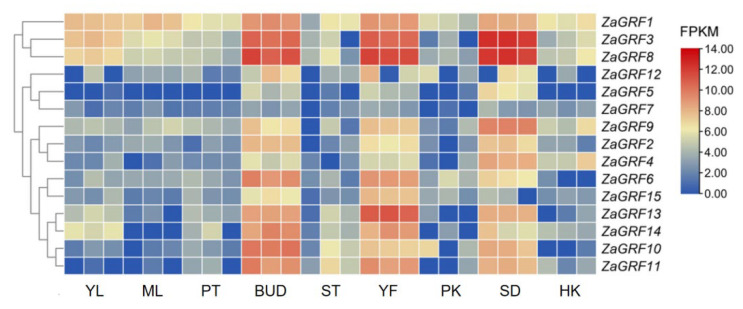
Tissue-specific expression pattern of *ZaGRF* genes. The heatmap was generated based on in silico analysis of the tissue-specific expression data of *ZaGRF* genes from the SRA database, and normalized log2 transformed values were used with hierarchical clustering. The transition from blue to red represents different expression levels. YL: young leaf; ML: mature leaf; PT petiole; BUD: terminal bud; STEM: stem; YF: young flower; PK: prick; SEED: seed; HK: husk.

**Figure 8 ijms-23-09043-f008:**
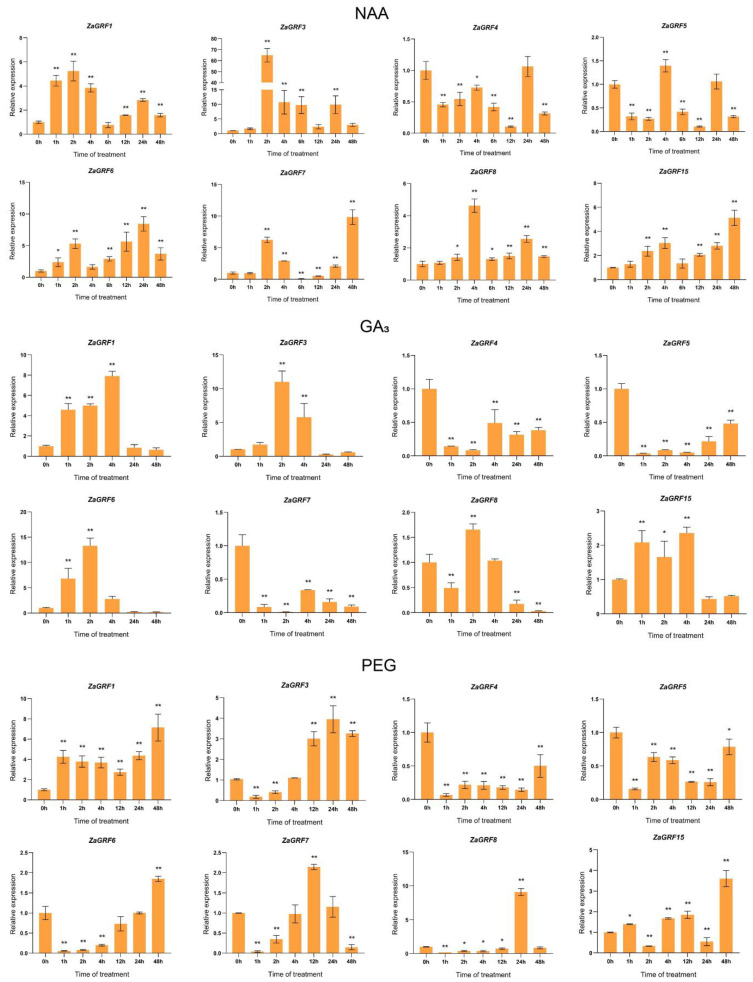
Expression of *ZaGRF* genes in response to NAA, GA_3_, and PEG treatment. The error bars represent the standard error of the means of three independent replicates of qRT-PCR analysis. *p* values were determined according to one-factor ANOVA test (* *p* < 0.05, ** *p* < 0.01).

**Figure 9 ijms-23-09043-f009:**
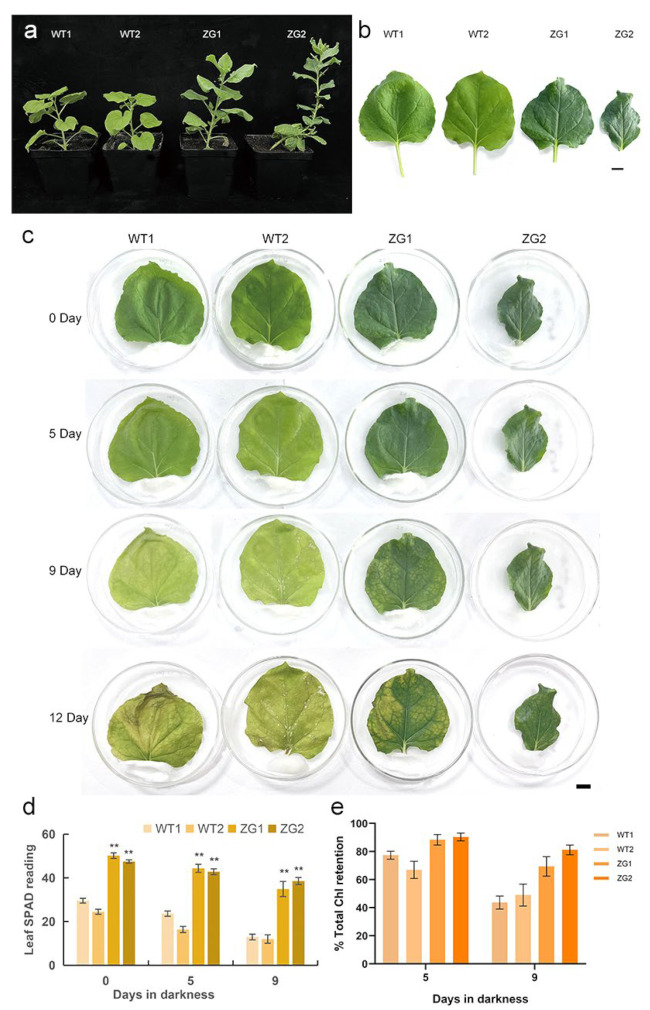
Phynotypic difference of WT and transgenic *N. benthamiana* overexpressing *ZaGRF6*. (**a**) Plant architecture; (**b**) Leaf phenotype; (**c**) Excised leaves were incubated in darkness for 0, 5, 9, and 12 days; (**d**) The chlorophyll content of leaves of transgenic and WT plants incubated in darkness for 0, 5, and 9 days; (**e**) Total chlorophyll retention of leaves of transgenic and WT plants after 5 and 9 days incubated in darkness. The error bars represent the standard error of the means of three independent replicates of qRT-PCR analysis. *p* values were determined according to one-factor ANOVA test (** *p* < 0.01) (*n* = 10).

**Figure 10 ijms-23-09043-f010:**
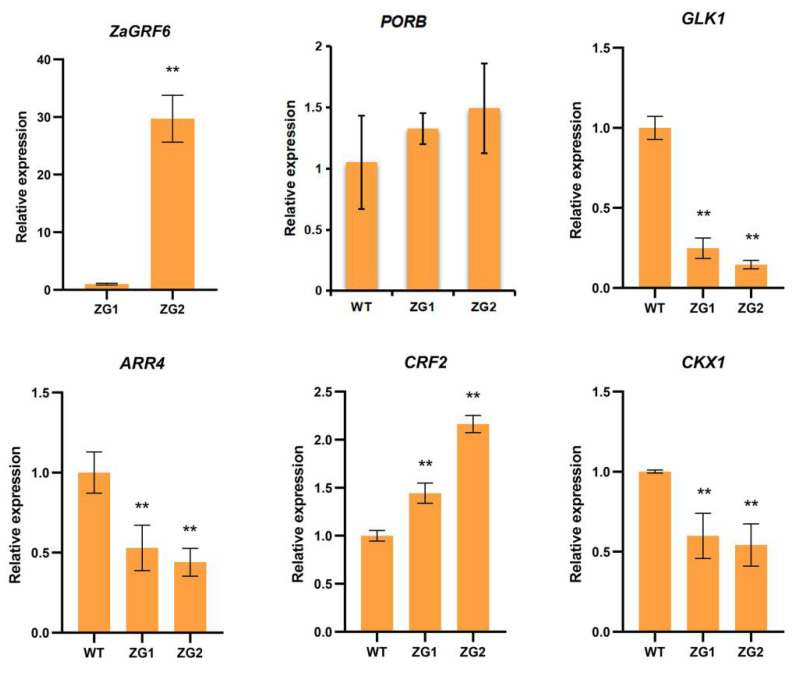
Expression levels of *ZaGRF6*, *PROB*, *GLK1*, *ARR4*, *CRF2*, and *CKX1* in transgenic and wild-type plants. The error bars represent the standard error of the means of three independent replicates of qRT-PCR analysis. *p* values were determined according to one-factor ANOVA test (** *p* < 0.01) (*n* = 3).

**Table 1 ijms-23-09043-t001:** Protein features of GRFs of *Z. armatum*.

Gene	Gene ID	Gene Locus	Start/Stop Codon	No. of Amino Acids	*M*_W_/kD	PI	Instability Index
*ZaGRF1*	Zardc16696.t1	Chr11	13,736,257~13,739,667	355	39.93	9.35	53.17
*ZaGRF2*	Zardc21503.t1	Chr15	3,863,606~3,866,118	508	55.39	9.15	40.83
*ZaGRF3*	Zardc26531.t1	Chr18	73,130,360~73,133,975	498	53.65	9.39	57.36
*ZaGRF4*	Zardc29111.t1	Chr20	29,291,029~29,292,832	384	42.22	8.30	59.43
*ZaGRF5*	Zardc30654.t1	Chr21	65,349,679~65,351,091	284	31.46	9.15	54.91
*ZaGRF6*	Zardc37188.t1	Chr26	32,600,857~32,604,386	333	36.81	8.66	63.71
*ZaGRF7*	Zardc37898.t1	Chr27	5,546,773~5,549,425	293	32.88	9.90	50.78
*ZaGRF8*	Zardc42395.t1	Chr31	51,746,517~51,750,356	652	70.82	6.24	47.54
*ZaGRF9*	Zardc45509.t1	Unanchored 456	284,498~286,320	384	42.28	8.60	59.09
*ZaGRF10*	Zardc49103.t1	Unanchored 5139	19,631~21,787	381	41.88	8.23	57.88
*ZaGRF11*	Zardc50859.t1	Unanchored 7776	8700~10,505	391	43.07	7.76	59.18
*ZaGRF12*	Zardc52135.t1	Unanchored 9706	24,661~26,568	474	51.70	7.24	58.53
*ZaGRF13*	Zardc52753.t1	Unanchored 10725	65,943~67,423	272	31.14	8.75	70.16
*ZaGRF14*	Zardc53033.t1	Unanchored 11223	3049~5043	503	55.15	6.98	57.08
*ZaGRF15*	Zardc54515.t1	Unanchored 13411	6765~8221	327	36.87	8.79	59.05

PI: isoelectric point; *M*_W_: molecular weight.

## Data Availability

Not applicable.

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
