# Peer review of "Genome-Wide Identification and Analysis of the Growth-Regulating Factor Family in Zanthoxylum armatum DC and Functional Analysis of ZaGRF6 in Leaf Size and Longevity Regulation"

_ijms, 2022, doi:10.3390/ijms23169043_

Round 1

Reviewer 1 Report

Most of the work was done in situ. However, this does not in any way reduce its value, but rather makes it more interesting. However, some aspects of research on protein structure, subcellular localization prediction, chromosome localizations, synteny analysis, etc. seem to be not sufficiently discussed.

Other Comments

Line 32: Abbreviations QLQ and WRC need to write the full name when meeting for the first time.

Line 65-66: It remains unclear why 15 of the 20 identified genes were obtained.

Line 95: Since the result was obtained only in situ, it is more correct to say that “can be localized” or “presumably localized”

Line 96:  “presumably localized”

Line 142: What does not anchored on chromosomes mean?

Line 190: Why and under what conditions 8 out of 15 analyzed genes were selected for further analysis?

Line 208: By what criteria was gene 6 selected for overexpression analysis?

Line 374-376: “Transcriptome 374 data for 9 different tissues (young leaf, mature leaf, petiole, terminal bud, stem, young 375 flower, prick, seed and husk) were downloaded from the NCBI database (PRJNA721257)” repeats line 377-378.

Line 377 “Heatmaps were drawn with TBtools software.” repeats line 383.

Author Response

We have responded one by one, please check the attachment.

Reviewer 2 Report

Comments and Suggestions for Authors

The authors identified GRFs members in Zanthoxylum armatum, did tissue-specific expression analysis, RT-qPCR, and the Tobacco instantaneous transformation test. The aims of study are clearer to understand. Some related references should be added, and the introduction and methods need to be improved.

Specific comments

Question 1

In line 17, "most of the ZaGRFs were highly expressed" need to add more details.

Question 2

In line 17, " most of selected ZaGRF genes" need to add more details.

Question 3

In line 25, "growth and development and stress resistance" should be "growth, development and stress resistance".

Question 4

The introduction was not well written. Authors should add the contents about GRFs response to auxin, gibberellin or drought studies.

Question 5

In line 51-55, the main purpose of this paper was not well introduced. Also, related references should be added.

Question 6

In line 58-62 More details need to be added. This paragraph should be well written.

Question 7

In line 65, GRFs family characteristics, according to what, comparative genomics or other methods, ZaGRFs was identified. The critical value was missing here and, in the M&M.

Question 8

In line 91 Table 2, Units of predicted physical location should be added.

Question 9

In line 103, it is better to list Table 3 as a supplemental table.

Question 10

In line116, supplemental table needs to present the members of GRFs in Z. armatum. and three other species.

Question 11

In line164, it is better to list Figure 6 as a supplemental figure.

Question 12

In line 167, for the tissue-specific expression of ZaGRF Genes, the authors should add the results of DEGs when comparing them with young flower or other tissues, (p-valve/q-value and log2FC). Also, the number of genes which were up regulated should be added. This paragraph needs to be improved. The references where the data were from should be added unless it was sequenced by the authors.

 Question 13

In line 190, there are 15 genes in this family. Why do the authors choose 8 to conduct the qRT-PCR analysis?

Question 14

Figure 9 and Figure 10 should be integrated into one figure.

Question 15

In line 247, the genes relative expression in Figure 11 was not unified.

Question 16

In the M&M the authors should add the analysis parameters and reference, e.g., in the 4.2 and 4.4.

Question 17

In line 408 The size of PCR product should be added in Table S1.

Round 2

Reviewer 1 Report

The manuscript can be considered for publication

Reviewer 2 Report

I have no more questions. Thanks!